# Bank-firm relationships and corporate ESG greenwashing

Hongyu Liu *

School of management,Guangzhou College of Technology and Business, Foshan, China

* liuhongyu@gzgs.edu.cn

## Abstract

This study explores how bank-firm relationships and executives with banking backgrounds influence greenwashing among Chinese A-share listed firms, based on Institutional and Agency Theories. Firms with bank shareholding or banking-experienced executives show a higher propensity for greenwashing, driven by executive compensation structures tied to inflated ESG scores. Financialization amplifies symbolic ESG disclosures, while transparency reduces greenwashing. Heterogeneity analysis reveals that non-state-owned enterprises and firms in less financially developed regions are more affected. These findings highlight the need for robust governance and transparency to promote authentic ESG practices, offering insights into mitigating greenwashing in emerging markets.

## 1. Introduction

With the growing global emphasis on sustainable development, corporate environmental, social, and governance (ESG) performance has become a focal point for regulators, investors, and the public. Governments around the world have implemented increasingly stringent environmental regulations and ESG disclosure requirements, while investors are integrating ESG criteria into capital allocation decisions. However, genuine ESG engagement often entails high costs, long-term investment horizons, and substantial operational adjustments. In response to these pressures, some firms resort to greenwashing-the strategic misrepresentation or exaggeration of ESG performance to project a more sustainable image without corresponding substantive action. Such behavior not only misleads stakeholders but also erodes the credibility of ESG initiatives and impedes real progress toward sustainability goals.

A growing body of literature has explored the antecedents of greenwashing. Most studies focus on external drivers, such as government regulations (Sun & Zhang, 2019 [1]; Hu et al., 2023 [2]), media scrutiny (Pope et al., 2016) [3], and NGO activism (Kim & Lyon, 2015) [4], or on internal firm-level factors, including size, profitability, governance structure, and financial constraints (Delmas & Burbano, 2011 [5]; Zhang, 2022 [6]). Yet one important internal governance factor-bank-firm relationships-has received surprisingly limited attention in the context of ESG misrepresentation.

**Data availability statement:** All relevant data are within the paper and its Supporting Information files.

**Funding:** This research was funded by Innovation Team Project for Ordinary Universities in Guangdong Province (Humanities and Social Sciences) (2023WCXTD026), Major Scientific Research Projects in Colleges and Universities in Guangdong (Grant No. 2021ZDJS122)(Grant No. 2022ZDJS142), and Guangdong Province Philosophy and Social Science Planning Project (GD23YGL28) (GD24CGL26). Funders were involved in the design of the study, data collection, analysis, interpretation of data, and writing of the manuscript.

**Competing interests:** The authors have declared that no competing interests exist.

Existing research on bank-firm relationships predominantly examines traditional lending channels, investigating how bank loans and debt covenants influence corporate ESG performance (Houston & Shan, 2022 [7]; Wang, 2023 [8]). However, banks can also be involved with firms through non-lending mechanisms, such as equity holdings or executive affiliations, which may shape ESG disclosure practices in more subtle but equally powerful ways. For example, when a bank becomes a shareholder or when corporate executives have prior experience in the banking industry, firms may face indirect pressures to align their ESG narratives with institutional investor expectations-potentially leading to symbolic compliance or greenwashing.

On the one hand, such relationships may enhance transparency by reducing information asymmetry and strengthening monitoring (Diamond, 1984 [9]; Mahrt-Smith, 2006 [10]). On the other hand, especially in emerging markets characterized by regulatory gaps and weaker enforcement, these relationships may foster rent-seeking behavior or strategic ESG signaling (Weinstein & Yafeh, 1998 [11]; Luo et al., 2011 [12]). For instance, in the Chinese context, studies have shown that bank ownership can weaken corporate governance (Lin et al., 2015 [13]) and distort executive incentives, but few have linked these dynamics directly to ESG greenwashing behavior. This gap presents a critical opportunity to examine whether non-lending bank-firm ties encourage firms to pursue symbolic ESG disclosure as a strategic response to institutional and financial pressures.

This study aims to investigate how bank shareholding and executives with banking backgrounds influence corporate greenwashing among Chinese A-share listed firms. We address three key research questions:

(1) Do non-lending bank-firm relationships increase the likelihood of greenwashing behavior?

(2) Through what mechanisms-such as executive compensation structures or corporate financialization—do these relationships operate?

(3) Under what conditions (e.g., ownership type, regional financial development, industry competition) are these effects more or less pronounced?

This study makes several important contributions to the literature
First, it extends greenwashing research by introducing a novel perspective on the role of institutional financial stakeholders-specifically banks-in shaping symbolic ESG disclosure via non-credit-based connections. While prior studies emphasize external regulation or market scrutiny, we show how internal financial affiliations, such as equity ties and executive networks, function as underappreciated channels of ESG influence.

Second, we enrich the understanding of bank-firm relationships by shifting the focus from their creditor role to a more comprehensive view that includes ownership structures and executive career pathways. This enables us to capture governance and incentive dynamics that are often overlooked in ESG-related studies and to explore how banks indirectly steer disclosure behavior in emerging institutional environments.

Third, this study makes a theoretical contribution by integrating Institutional Theory, Agency Theory, and Legitimacy Theory to explain the drivers of ESG misrepresentation. Beyond this integrative framework, we delve into causal mechanisms, particularly the mediating role of executive compensation and the moderating roles of financialization and narrative tone in ESG reports. We also explore alternative explanations and competing pathways—such as industry characteristics and regional financial development-that may influence the observed relationships. This depth of mechanism analysis responds directly to calls for more nuanced exploration of how and why symbolic ESG behavior emerges.

Fourth, we construct a multi-source greenwashing index based on discrepancies across three leading ESG rating systems (CSMAR, Huazheng, and Bloomberg), allowing for more robust measurement than single-source proxies. By combining textual, financial, and governance data, our empirical approach offers stronger inference and broader applicability to ESG research and policy evaluation.

The remainder of this paper is organized as follows: Section 2 presents the literature review and hypothesis development. Section 3 introduces the data, variables, and empirical strategy. Section 4 reports the main findings and robustness checks. Section 5 discusses implications and concludes.

## 2. Literature review and hypotheses development

### 2.1. Literature review

Research on corporate greenwashing suggests that firms engage in ESG misrepresentation due to a combination of external and internal factors. Delmas and Burbano (2011) [5] classify greenwashing drivers into four categories: non-market external factors, market external factors, organizational factors, and individual psychological factors. Market external factors, such as peer pressure, encourage firms to mimic their competitors' ESG strategies, often leading to exaggerated or misleading sustainability claims (Marquis et al., 2016) [14]. Organizational factors, such as misaligned incentive structures, can also contribute to greenwashing-when managerial compensation is tied to ESG disclosures rather than substantive sustainability performance, firms may prioritize superficial compliance over actual ESG improvements (Moser & Martin, 2012) [15].

At the firm level, research has identified financial constraints as a key driver of greenwashing. Kim and Lyon (2015) [4] demonstrate that rapidly growing firms, especially in capital-intensive industries, often struggle to balance operational efficiency with regulatory compliance, making them more prone to ESG misrepresentation. Zhang (2022) [6] further argues that firms facing financial distress may resort to greenwashing as a cost-effective strategy to attract ESG-conscious investors without incurring the expenses associated with genuine sustainability initiatives. Similarly, Walker and Wan (2012) [16] highlight how firms under financial strain tend to selectively disclose ESG efforts to enhance their market reputation, even when their actual sustainability performance remains subpar.

The role of financial institutions, particularly banks, in shaping corporate ESG behavior has been relatively underexplored. Existing studies suggest that banks influence firms' ESG performance primarily through lending relationships. Houston and Shan (2022) [7] argue that banks can promote ESG transparency by incorporating ESG covenants into loan agreements or offering lower interest rates for sustainable firms. Similarly, Wang (2023) [8] finds that borrowing firms tend to align their ESG disclosures with lender expectations to secure financing. However, these studies largely focus on banks' creditor role, overlooking the potential influence of banks as equity holders or through executive affiliations.

This study addresses this gap by investigating how bank shareholding and executives with banking backgrounds influence corporate greenwashing. Unlike lending relationships, bank equity ownership grants banks direct influence over corporate governance and decision-making. On the one hand, bank ownership may reduce information asymmetry, providing firms with better access to capital and encouraging responsible ESG practices (Diamond, 1984) [9]. On the other hand, banks as shareholders may prioritize short-term financial performance, incentivizing firms to engage in symbolic ESG disclosures rather than substantive sustainability efforts (Weinstein & Yafeh, 1998 [11]; Agarwal & Elston, 2001 [17]).

Executives with banking backgrounds may further intensify greenwashing by leveraging their financial expertise to manage ESG perceptions strategically. Dittmann et al. (2010) [18] suggest that bank-affiliated executives may enhance firms' access to capital but may also foster a short-term, finance-driven approach to ESG. Lyon and Montgomery (2015) [19] argue that such executives may use ESG disclosures as a tool for managing investor expectations, rather than as a reflection of genuine sustainability performance. In emerging markets like China, where regulatory oversight remains weak and information asymmetry is high, bank-affiliated executives may have both the incentive and expertise to selectively disclose ESG information to appeal to investors, thus increasing the likelihood of greenwashing.

While previous research highlights how banks influence ESG performance as creditors, it remains unclear how bank ownership and executive affiliations affect corporate greenwashing. This study contributes to the literature by exploring these alternative channels, shedding light on the complex and often conflicting roles that banks play in shaping corporate ESG narratives.

## 2.2 Theoretical framework

To develop our hypotheses, we adopt three key theoretical perspectives: Institutional Theory, Agency Theory, and Legitimacy Theory.

Institutional Theory suggests that organizations conform to external pressures-such as regulations, market expectations, and investor preferences-to gain legitimacy and maintain competitive advantage (DiMaggio & Powell, 1983) [20]. Banks, as institutional investors, face increasing pressure from regulators, socially responsible investors, and the public to support ESG-aligned firms. Consequently, firms with bank shareholders may feel compelled to enhance their ESG disclosures, even if their actual sustainability efforts do not match their reporting. Recent studies indicate that firms in markets undergoing capital liberalization strategically use greenwashing to attract foreign and institutional investors, particularly when they face high financial constraints (Liu et al., 2024) [21].

Agency Theory emphasizes the information asymmetry between managers and shareholders, leading to potential conflicts of interest. When banks become shareholders, they participate in corporate governance, yet may lack full visibility into firms' actual ESG performance. In this context, firms may use symbolic ESG disclosures to satisfy shareholder expectations while avoiding substantive sustainability commitments (Lin et al., 2015 [13]; Luo et al., 2011 [12]). This effect is especially pronounced in emerging markets, where weak regulatory enforcement allows firms to engage in ESG misrepresentation without significant consequences.

Legitimacy Theory posits that firms engage in behaviors that align with societal expectations to maintain legitimacy and secure financial support (Suchman, 1995) [22]. Executives with banking backgrounds, given their experience in financial markets, may be particularly attuned to external legitimacy pressures. Research suggests that firms with strong institutional affiliations are more likely to engage in ESG reporting strategies aimed at managing investor perceptions rather than achieving genuine sustainability goals (Kim & Lyon, 2015) [4]. In less regulated markets, these executives may leverage greenwashing as a low-cost strategy to enhance corporate reputation while minimizing the financial burden of substantive ESG investments (Zhang, 2022) [6].

## 2.3 Hypotheses development

Bank shareholding increases the likelihood of corporate greenwashing. Bank shareholding represents a strategic relationship that goes beyond traditional lending and positions banks as stakeholders with a vested interest in corporate performance and governance (Degryse et al., 2021) [23]. According to Institutional Theory, organizations in highly regulated and normatively pressured environments often adopt practices aligned with social expectations to gain legitimacy and acceptance (DiMaggio & Powell, 1983) [20]. Banks, as major institutional investors, face significant external pressure from regulatory bodies, socially conscious investors, and the public to ensure their investments align with environmental, social, and governance (ESG) standards. Thus, when banks become shareholders, their institutional influence can significantly shape

corporate behavior. Firms seeking continued support from bank shareholders may feel compelled to enhance their ESG disclosures to align with banks' preferences for "green" investments, even if these do not fully reflect their actual sustainability practices. Recent research indicates that capital market liberalization encourages firms to engage in ESG reporting greenwashing to attract foreign investors, with effects more pronounced among firms facing high financing constraints(Liu et al., 2024) [21]. This pressure to conform to institutional expectations leads firms to prioritize image-enhancing ESG disclosures, often without substantive changes, to gain legitimacy in the marketplace (Huang & Kong, 2020) [24].

Agency Theory further supports this hypothesis by emphasizing the impact of information asymmetry between shareholders (in this case, banks) and managers. Although banks as shareholders participate in corporate governance, they may lack full visibility into a firm's actual ESG performance, creating an incentive for managers to focus on symbolic ESG disclosures rather than substantive improvements. In emerging markets, where financial systems are less developed, bank-firm relationships often exert even greater influence over firms' strategies. Firms may opt for symbolic rather than substantive ESG enhancements to satisfy shareholder expectations while minimizing scrutiny. By relying on greenwashing, these firms can attract or retain bank support while presenting an image of ESG compliance (Lin et al., 2015 [13]; Luo et al., 2011 [12]). Based on this analysis, the following hypothesis is proposed:

H1: Bank shareholding increases the likelihood of corporate greenwashing.

Firms with executives who have banking backgrounds have a higher propensity for greenwashing. Agency Theory suggests that executives' backgrounds and experiences shape their decision-making processes, influencing the alignment of interests between agents (executives) and principals (shareholders). Executives with banking backgrounds may be more inclined to employ symbolic practices, such as exaggerated ESG disclosures, to satisfy investors' expectations for corporate social responsibility (Dittmann et al., 2010) [18]. Studies show that the adoption of circular economy models has led to broader "ESG-washing" practices, where firms focus more on managing stakeholder perceptions than on substantive sustainability efforts (Todaro & Torelli, 2024) [25]. These executives often have a nuanced understanding of capital markets and investor expectations, which enables them to skillfully use information management strategies to elevate the company's standing in financial markets. Banking-background executives may leverage their financial expertise to selectively disclose or even inflate ESG performance, thus attracting investment even if these disclosures do not accurately represent the firm's ESG practices (Weinstein & Yafeh, 1998 [11]; Agarwal & Elston, 2001 [17]). Legitimacy Theory also provides support for this hypothesis. Executives with banking backgrounds tend to be more attuned to external assessments of corporate compliance and legitimacy due to their familiarity with the financial sector's demands. To ensure favorable perceptions from investors and other stakeholders, these executives may strategically enhance the firm's ESG image through selective disclosures that align with societal and market expectations, effectively leveraging greenwashing to build legitimacy and attract capital (Kim & Lyon, 2015) [4]. In emerging markets with relatively lax ESG regulations, these executives may be even more incentivized to rely on greenwashing as a low-cost strategy to boost the firm's attractiveness and market value without the financial burdens of implementing genuine ESG improvements (Zhang, 2022) [6]. Greenwashing behavior effectively constructs a unique "ESG signaling amplifier" mechanism. Although such behavior is difficult to detect, it conveys more visually appealing and readily interpretable ESG information to investors and creditors, thereby enhancing the readability and perceived clarity of sustainability disclosures (Marquis et al., 2016) [14]. This can lead institutional investors to develop optimistic biases in their ESG evaluations, raising their expectations of the firm's future behavior and performance (Gatti et al., 2021) [26], ultimately resulting in lower financing costs and higher firm valuation (Merve et al., 2020) [27]. Moreover, consumers tend to exhibit a preference for green-labeled products (Lee et al., 2018) [28], but often lack the ability to distinguish genuine sustainability efforts from greenwashing. This allows firms to boost sales revenues by leveraging ESG claims to market "green" products more effectively (Chen & Dagestani, 2023) [29]. In addition, the enhanced green image generated through greenwashing may improve a firm's access to fiscal and financial support, including government subsidies, bank loans, and tax incentives (Zhang et al., 2020) [30]. These supports help alleviate financing constraints and promote business development, which

in turn contributes to greater firm value (Qilin et al., 2022 [31]; Chen et al., 2023 [29]). Based on this analysis, the following hypothesis is proposed:

H2: Firms with executives who have banking backgrounds have a higher propensity for greenwashing.

## 3. Research design

### 3.1. Formatting of mathematical components

The data sample for this study consists of A-share listed companies in China from 2010 to 2023. Firms are selected based on the availability of Environmental, Social, and Governance (ESG) data from both the CSMAR and Huazheng databases, with an additional check on Bloomberg data to ensure the robustness of greenwashing measurements. The primary sample includes firms for which normalized CSR scores from these databases are available, enabling a comprehensive assessment of greenwashing behavior defined as the discrepancy between firms' disclosed and actual ESG performance. During the sample selection, companies that were delisted or newly listed with less than one year of available data were excluded. Further, observations with extreme outliers or incomplete information for key variables were removed to maintain data consistency and reliability.

After these filtering criteria, the final sample includes 40486 firm-year observations from 4454 companies. To mitigate the impact of outliers, a 1% winsorization was applied to continuous variables at both tails of the distribution. This trimming process aims to enhance the robustness of the regression results by reducing the influence of extreme values on the study's findings.

### 3.2. Variable definition

In this paper, the degree of corporate greenwashing serves as the dependent variable. Following prior studies, greenwashing behavior is measured as the discrepancy between a firm's disclosed Environmental, Social, and Governance (ESG) score and its actual ESG performance, which is calculated based on standardized CSR scores from CSMAR and Huazheng databases, as well as Bloomberg scores (Long et al., 2024) [32]. Specifically, we construct three measures of greenwashing behavior: (1) the difference between the standardized CSMAR CSR score and the Huazheng ESG comprehensive score; (2) the difference between the standardized CSMAR CSR score and the Huazheng ESG comprehensive rating; and (3) the difference between the standardized Bloomberg ESG score and the Huazheng ESG comprehensive score. We define the firm's greenwashing index as the discrepancy between its ESG disclosure performance and actual ESG behavior, based on peer-adjusted standardized scores. Following prior literature, we calculate greenwashing as the difference between the firm's percentile rank in ESG disclosure scores (from CSMAR and Bloomberg) and its percentile rank in ESG performance scores (based on Huazheng ratings). A positive greenwashing score indicates that a firm overstates its sustainability efforts, potentially misleading stakeholders with inflated ESG disclosures relative to its actual ESG performance. The use of multiple data sources enhances the robustness and objectivity of our greenwashing measure.

The main explanatory variable in this study is the bank-firm relationship. This variable takes a value of 1 if a firm holds bank shares, if a bank holds shares in the firm, or if one or more executives have a banking background, and 0 otherwise. This variable captures the influence of bank affiliation on corporate governance and disclosure practices, especially regarding greenwashing behaviors (Zhai et al., 2014) [33].

Following established research, several firm-level characteristics are included as control variables. These include Firm Size (Size), measured as the natural logarithm of total assets; Return on Assets (ROA), defined as net profit divided by total assets; Growth, calculated as the revenue growth rate; Leverage (Lev), defined as total liabilities divided by total assets; Fixed Asset Ratio (FAR), calculated as fixed assets divided by total assets; Board Independence (Indep), defined as the proportion of independent directors on the board; Firm Age (FirmAge), measured as the natural logarithm of years since establishment plus one; Board Size (Board), the natural logarithm of the number of

board directors; Institutional Ownership (Inst), defined as the proportion of shares held by institutional investors; Operating Cash Flow (OCF), calculated as net cash flow from operating activities divided by revenue; and Book-to-Market Ratio (BM), defined as the book value of equity divided by the market value of equity (Yu et al., 2020 [34]; Zhang, 2022 [6]; Pope et al., 2024 [35]).

In addition, several other variables are introduced to explore mechanisms and moderating effects. Executive Compensation (Msalary) is measured as the natural logarithm of the average compensation of executives plus one. Positive Tone (TONE) in annual reports is calculated as the ratio of the difference between the number of positive and negative words over the total word count, reflecting managerial optimism (Liu et al., 2023) [36]. Financialization Ratio (FINRATIO), representing the degree of corporate financialization, is defined as the ratio of financial assets to total assets (Gong et al., 2023) [37]. Financial Agglomeration (FA) measures the concentration of financial industry employment in a city relative to the national average, capturing regional financial development (Xie et al., 2021) [38]. Market Competition is assessed using the Herfindahl-Hirschman Index (HHI), calculated as the sum of squared market shares of companies within the same industry, indicating the level of industry competition (Pope et al., 2024 [35]).

The definitions and measurement methods for each variable are summarized in Table 1.

### 3.3. Model construction

To examine the impact of bank-firm relationships and executive banking backgrounds on corporate greenwashing behavior, we use the following baseline regression model:

$$Greenwash_{it} = \alpha + \beta_1 BankRel_{it} + \gamma X_{it} + \lambda_t + \eta_i + \varepsilon_{it}$$

Where:

$Greenwash_{it}$ is the greenwashing level of firm $i$ in year $t$, measured by the discrepancy between standardized ESG disclosure scores and actual ESG performance scores.

$BankRel_{it}$ is the main independent variable, representing the bank-firm relationship. It takes a value of 1 if the firm holds bank shares, the bank holds firm shares, or if any executives havea banking background; otherwise, it is 0.

$X_{it}$ is a vector of control variables, including firm size (Size), return on assets (ROA), growth (Growth), leverage (Lev), fixed asset ratio (FAR), board independence (Indep), firm age (FirmAge), board size (Board), institutional ownership (Inst), operating cash flow (OCF), and book-to-market ratio (BM).

$\lambda_t$ and $\eta_i$ are year and firm fixed effects, respectively, used to control for unobserved heterogeneity across time and firms.

$\varepsilon_{it}$ is the error term.

To further explore potential moderating effects of variables such as Positive Tone and Financialization Ratio on the bank-firm relationship's impact on greenwashing, we extend the model with interaction terms as follows:

$$Greenwash_{it} = \alpha + \beta_1 BankRel_{it} + \beta_2 ModVar_{it} + \beta_3 (BankRel_{it} \times ModVar_{it}) + \gamma X_{it} + \lambda_t + \eta_i + \varepsilon_{it}$$

Where:

$ModVar_{it}$ represents the moderating variables, such as Positive Tone (TONE) Or Financialization Ratio (FINRATIO).

$BankRel_{it} \times ModVar_{it}$ is the interaction term, capturing how the impact of the bank-firm relationship on greenwashing changes with variations in the moderating variable.

This regression model allows us to test whether bank-firm relationships and executive banking backgrounds significantly contribute to greenwashing behavior, while the interaction terms help determine whether factors like positive tone or financialization intensify or mitigate this effect.

**Table 1. Variable Definition.**

| Type | Variables | Definition and Measurement |
|---|---|---|
| Dependent Variable | Green Washing Level 1 | Difference between standardized CSR score from the CSMAR database and Huazheng standardized ESG Integral Score |
| | Green Washing Level 2 | Difference between standardized CSR score from the CSMAR database – Huazheng Standardized ESG Integral Rating |
| | Green Washing Level 3 | Difference between standardized ESG score from the Bloomberg database and Huazheng Standardized ESG Integral Score |
| Independent Variable | Bank-Firm Relationship | The company holds bank shares, the bank holds company shares, or executives have a banking background. If any condition is met, it takes 1; otherwise, it takes 0. |
| Control Variables | Size | Natural logarithm of total assets |
| | ROA | Net profit/ total assets |
| | Growth | Revenue growth rate |
| | Lev | Total liabilities/ total assets |
| | FAR | Fixed assets/ total assets |
| | Indep | Number of independent directors/ total number of board directors |
| | Firm Age | Natural logarithm of the number of years since establishment plus 1 |
| | Board | Natural logarithm of the number of board directors |
| | Inst | Number of shares held by institutional investors/ total share capital |
| | OCF | Net cash flow from operating activities/ revenue |
| | BM | Book value of shareholders' equity/ market value |
| Other Variables | Msalary | Natural logarithm of average executive compensation plus 1 |
| | TONE | (Number of positive words – number of negative words)/ (number of positive words + number of negative words) |
| | FINRATIO | Financial assets/ total assets |
| | FA | (Number of employees in the financial industry in the prefecture-level city/ number of employees in the prefecture-level city)/ (number of employees in the financial industry nationwide/ number of employees nationwide) |
| | HHI | Sum of squares of the ratio of individual company's revenue to industry revenue |

## 4. Further Analysis

### 4.1 Baseline Regression Results

Table 2 displays the baseline regression results examining the impact of bank-firm relationships on corporate greenwashing behavior, specifically within Chinese A-share listed firms. The dependent variable, Greenwashing Level, quantifies the discrepancy between a firm's disclosed ESG score and its actual ESG performance. Column (1) incorporates only firm and year fixed effects to control for constant firm-level characteristics and yearly variations, while Column (2) further introduces industry-year fixed effects to account for industry-specific annual fluctuations. Columns (3) and (4) extend the model by adding various firm-level control variables, including Firm Size, ROA, and Leverage, to better isolate the effect of bank-firm relationships on greenwashing.

Across all specifications, the coefficient for Bank-Firm Relationship is positive and statistically significant, indicating that firms with closer bank affiliations are more likely to engage in greenwashing behavior. This finding supports Hypothesis 1, suggesting that firms with banking relationships may be motivated to inflate their ESG disclosures to align with banks' preferences for sustainable investment profiles and to appeal to ESG-conscious investors.

### 4.2 Intermediary Effect Mechanism

Table 3 investigates the mediating role of executive compensation in the relationship between bank-firm affiliations and greenwashing. The mediating variable, Msalary, represents the natural logarithm of average executive compensation

**Table 2. Baseline Regression Results.**

| | (1) | (2) | (3) | (4) |
| --- | --- | --- | --- | --- |
| | Green Washing Level | Green Washing Level | Green Washing Level | Green Washing Level |
| Banking-frim relationship | 0.045** | 0.047*** | 0.045** | 0.046*** |
| | (2.504) | (3.132) | (2.692) | (3.342) |
| Size | | | −0.404*** | −0.412*** |
| | | | (−14.911) | (−16.383) |
| ROA | | | −0.663*** | −0.655*** |
| | | | (−9.211) | (−8.969) |
| Growth | | | 0.087*** | 0.084*** |
| | | | (3.175) | (3.085) |
| Lev | | | 0.828*** | 0.875*** |
| | | | (16.122) | (17.532) |
| FAR | | | 0.026 | 0.048 |
| | | | (0.281) | (0.583) |
| Indep | | | −1.364*** | −1.321*** |
| | | | (−2.988) | (−2.964) |
| Board | | | −0.156 | −0.145 |
| | | | (−1.355) | (−1.282) |
| FirmAge | | | −0.213 | 0.032 |
| | | | (−1.682) | (0.378) |
| BM | | | 0.311*** | 0.312*** |
| | | | (5.604) | (6.326) |
| OCF | | | 0.128 | 0.099 |
| | | | (1.085) | (1.012) |
| Inst | | | 0.080 | 0.092 |
| | | | (0.790) | (0.994) |
| _cons | −0.089*** | −0.089*** | 9.746*** | 9.146*** |
| | (−9.280) | (−19.099) | (28.999) | (26.336) |
| Firm FE | Yes | Yes | Yes | Yes |
| Year FE | Yes | No | Yes | No |
| Industry*Year FE | No | Yes | No | Yes |
| N | 37539 | 37529 | 37539 | 37529 |
| $R^2$ | 0.689 | 0.698 | 0.703 | 0.712 |
| Adj. $R^2$ | 0.651 | 0.658 | 0.666 | 0.674 |

and explores whether banking affiliations tie executive incentives to inflated ESG disclosures. This model aims to assess whether firms with bank affiliations provide higher executive compensation specifically tied to ESG scores, potentially driving greenwashing.

The results in Table 3 reveal a statistically significant positive interaction between Bank-Firm Relationship and Huazheng CSR Score on executive compensation. This finding supports Hypothesis 2, suggesting that firms with bank affiliations tend to structure executive compensation in alignment with ESG disclosures rather than actual ESG performance. This mediating role of executive compensation implies that banking relationships not only create pressure for enhanced ESG disclosure but also incentivize executives to prioritize symbolic ESG improvements over substantive sustainability initiatives.

**Table 3. Mechanism-Mediating-Salary.**

| | (1) |
| --- | --- |
| | **Msalary** |
| Banking-frim relationship | 0.026* |
| | (1.849) |
| Huazheng CSR Integral Score | 0.001 |
| | (0.879) |
| Banking-frim relationship*Huazheng CSR Integral Score | 0.011*** |
| | (4.136) |
| Size | 0.088*** |
| | (5.181) |
| ROA | 0.390*** |
| | (4.665) |
| Growth | −0.018 |
| | (−0.632) |
| Lev | −0.012 |
| | (−0.298) |
| FAR | −0.036 |
| | (−0.627) |
| Indep | 0.181 |
| | (1.343) |
| Board | 0.135* |
| | (1.995) |
| FirmAge | −0.144** |
| | (−2.409) |
| BM | −0.122*** |
| | (−3.025) |
| OCF | 0.082 |
| | (1.174) |
| Inst | 0.198** |
| | (2.218) |
| _cons | 8.247*** |
| | (22.264) |
| Firm FE | Yes |
| Industry*Year FE | Yes |
| N | 37529 |
| $R^2$ | 0.979 |
| Adj. $R^2$ | 0.976 |

t-Statistics calculated using standard errors clustered by industry and year are reported in parentheses. ***, **, and * denote statistical significance at the 1%, 5%, and 10% levels.

These results highlight the indirect effect of banking ties in promoting greenwashing by linking executive rewards to ESG performance metrics, thus driving a focus on disclosure rather than actual sustainability outcomes.

## 4.3. Test for moderating effects

A higher level of corporate financialization is found to positively moderate the relationship between bank-firm ties and greenwashing, thereby intensifying the likelihood of symbolic ESG disclosure. Financialized firms often prioritize short-term

financial performance, and close ties with banks provide them with expanded access to external financing. This resource availability increases their capacity to engage in greenwashing. To maintain stock prices or attract additional capital, such firms may be more inclined to exaggerate their environmental performance to appeal to investors or satisfy regulatory expectations. From another perspective, firms with high degrees of financialization may face greater pressure to meet short-term earnings targets. With access to funding via bank connections, these firms are more likely to adopt low-cost, high-return strategies-of which greenwashing is a prime example. Symbolic ESG disclosure offers a relatively inexpensive way to enhance corporate reputation, improve market perception, and secure financial support, potentially creating a self-reinforcing cycle. Moreover, high financialization may indicate that a firm has greater financial resources at its disposal, which can be used to support greenwashing activities such as purchasing environmental certifications, conducting promotional ESG campaigns, or inflating sustainability scores. In such cases, bank-firm relationships not only facilitate financing but may also provide implicit protection or leniency in oversight, thereby lowering the risk of greenwashing being detected. Collectively, these factors suggest that financialization exacerbates the enabling effect of bank affiliations on symbolic ESG behavior.

ESG disclosure transparency serves as an important mitigating factor in the positive relationship between bank-firm affiliations and corporate greenwashing behavior. On the one hand, the information-transmitting function of ESG ratings enhances corporate transparency and strengthens internal governance structures, thereby acting as an external constraint on opportunistic disclosure practices (Yu et al., 2020) [34]. On the other hand, enhanced ESG disclosure transparency helps to reduce signaling asymmetry, making it more difficult for firms to engage in strategic misrepresentation. As such, it is recognized as an effective mechanism for curbing "commercial greenwashing" behaviors (Wook et al., 2023) [39].

Table 4 examines the moderating effects of Positive Tone in annual reports and Financialization Ratio on the relationship between bank-firm affiliations and greenwashing. Positive Tone, calculated as the difference between positive and negative word counts in annual reports relative to total word count, captures management's optimism in ESG disclosures. Financialization Ratio, defined as the ratio of financial assets to total assets, indicates the level of corporate financialization.

The results show that Positive Tone has a negative moderating effect, weakening the impact of bank-firm relationships on greenwashing. This suggests that firms with more transparent and optimistic communication are less likely to engage in greenwashing despite bank affiliations. In contrast, a higher Financialization Ratio has a positive moderating effect, intensifying the association between bank-firm relationships and greenwashing. This implies that highly financialized firms may prioritize short-term financial performance over genuine ESG improvements, thereby heightening greenwashing tendencies in firms with banking affiliations. The findings underscore that transparency in ESG disclosures can mitigate greenwashing, while financialization amplifies it, particularly when bank relationships are present.t-Statistics calculated using standard errors clustered by industry and year are reported in parentheses. ***, **, and * denote statistical significance at the 1%, 5%, and 10% levels.

Table 5 presents a series of robustness checks designed to verify the consistency and reliability of the baseline regression results. The tests employ alternative specifications of the dependent variable, model assumptions, and sample restrictions to ensure the observed relationship between bank-firm relationships and greenwashing is not driven by model artifacts or sample selection bias.

Columns (1) and (2) test the sensitivity of results to different greenwashing measurement methods. Specifically, Column (1) constructs Greenwashing Level 1 as the difference between standardized ESG disclosure scores from CSMAR and ESG performance scores from Huazheng. Column (2), labeled Greenwashing Level 2, substitutes Bloomberg's ESG disclosure scores for Huazheng as a robustness check. In both cases, the coefficient on the Banking-Firm Relationship variable remains significantly positive (0.044 and 0.030, respectively), indicating that the positive link between bank ties and greenwashing is robust to changes in measurement methodology.

Columns (3) and (4) implement the Tobit model to account for potential censoring in the greenwashing variable, which is bounded below at zero. This model is particularly appropriate when the dependent variable is left-censored or has

**Table 4. Mechanism-Moderating.**

| | (1) | (1) |
|---|---|---|
| | **Green Washing Level** | **Green Washing Level** |
| Banking-frim relationship | 0.048*** | 0.047*** |
| | (3.394) | (3.391) |
| Positive Tone | −0.007*** | |
| | (−9.075) | |
| Banking-frim relationship*Positive Tone | −0.002* | |
| | (−2.013) | |
| Finratio | | 0.070 |
| | | (0.476) |
| Banking-frim relationship*Finratio | | 0.438** |
| | | (2.838) |
| Size | −0.397*** | −0.412*** |
| | (−15.942) | (−16.744) |
| ROA | −0.619*** | −0.654*** |
| | (−7.810) | (−8.794) |
| Growth | 0.088*** | 0.085*** |
| | (3.263) | (3.105) |
| Lev | 0.875*** | 0.878*** |
| | (18.036) | (17.901) |
| FAR | 0.025 | 0.054 |
| | (0.311) | (0.646) |
| Indep | −1.342*** | −1.317*** |
| | (−3.033) | (−2.935) |
| Board | −0.140 | −0.144 |
| | (−1.229) | (−1.277) |
| FirmAge | 0.059 | 0.028 |
| | (0.700) | (0.321) |
| BM | 0.302*** | 0.313*** |
| | (6.054) | (6.315) |
| OCF | 0.090 | 0.100 |
| | (0.933) | (1.035) |
| Inst | 0.098 | 0.094 |
| | (1.050) | (1.029) |
| _cons | 9.029*** | 9.153*** |
| | (25.651) | (26.800) |
| Firm FE | Yes | Yes |
| Industry*Year FE | Yes | Yes |
| N | 37529 | 37529 |
| $R^2$ | 0.713 | 0.712 |
| Adj. $R^2$ | 0.675 | 0.674 |

limited distributional range. Column (3) applies the Tobit model to the full greenwashing score, while Column (4) refines the specification by re-estimating the model with an alternative Tobit setting. In both columns, the coefficients on bank-firm relationships (0.079 and 0.069, respectively) remain positive and statistically significant at the 1% level, further confirming that the core findings are not driven by model misspecification due to truncation or skewed distributions.

**Table 5. Robustness Test.**

| | Change Green Washing Measurement | | Tobit | | PSM | |
|---|---|---|---|---|---|---|
| | (1) | (2) | (3) | (4) | (5) | (6) |
| | Green Washing Level 1 | Green Washing Level 2 | Green Washing Level | Green Washing Level | Let Green Washing Level=0 when it<0 | Delete Brown Washing Observation |
| Banking-frim relationship | 0.044*** | 0.030** | 0.079*** | 0.069*** | 0.028*** | 0.047*** |
| | (3.077) | (2.368) | (3.144) | (3.310) | (3.107) | (3.324) |
| Size | −0.405*** | −0.164*** | −0.497*** | −0.463*** | −0.204*** | −0.246*** |
| | (−16.658) | (−4.037) | (−18.341) | (−12.624) | (−13.868) | (−11.385) |
| ROA | −0.591*** | −0.523** | −2.995*** | −0.896*** | −0.536*** | −0.517*** |
| | (−9.049) | (−2.442) | (−12.098) | (−6.575) | (−6.979) | (−4.444) |
| Growth | 0.083*** | 0.086*** | 0.186*** | 0.083** | 0.046** | 0.047** |
| | (3.457) | (4.359) | (4.433) | (2.246) | (2.675) | (2.558) |
| Lev | 0.834*** | 0.668*** | 1.236*** | 1.005*** | 0.525*** | 0.722*** |
| | (14.549) | (10.188) | (18.699) | (15.473) | (12.541) | (12.474) |
| FAR | 0.040 | 0.408*** | −0.135 | −0.137 | 0.112 | 0.229** |
| | (0.398) | (3.756) | (−1.084) | (−1.303) | (1.632) | (2.362) |
| Indep | −1.372*** | −1.224*** | −1.589*** | −1.764*** | −0.508*** | −0.695*** |
| | (−3.199) | (−3.071) | (−4.103) | (−3.302) | (−3.821) | (−3.163) |
| Board | −0.149 | −0.027 | −0.279** | −0.210 | −0.070 | −0.119 |
| | (−1.389) | (−0.257) | (−2.155) | (−1.219) | (−1.581) | (−1.365) |
| FirmAge | 0.037 | 0.154 | 0.296*** | −0.125 | 0.269*** | 0.298** |
| | (0.429) | (0.819) | (4.550) | (−1.121) | (3.328) | (2.781) |
| BM | 0.305*** | 0.052 | 0.209*** | 0.366*** | 0.102*** | 0.067 |
| | (6.417) | (0.783) | (3.044) | (4.101) | (3.843) | (1.181) |
| OCF | 0.125 | 0.460*** | −0.418** | 0.145 | −0.034 | 0.018 |
| | (1.311) | (4.485) | (−2.256) | (1.211) | (−0.502) | (0.167) |
| Inst | 0.071 | −0.001 | 0.042 | 0.091 | −0.020 | −0.010 |
| | (0.717) | (−0.004) | (0.734) | (0.923) | (−0.457) | (−0.211) |
| _cons | 9.034*** | 3.204*** | 10.905*** | 11.204*** | 4.362*** | 5.869*** |
| | (26.603) | (7.450) | (19.382) | (20.688) | (11.009) | (15.813) |
| Firm FE | Yes | Yes | Yes | Yes | Yes | Yes |
| Industry*Year FE | Yes | Yes | Yes | Yes | Yes | Yes |
| N | 37529 | 13062 | 37902 | 21780 | 37529 | 17225 |
| $R^2$ | 0.697 | 0.676 | | 0.721 | 0.572 | 0.479 |
| Adj. $R^2$ | 0.657 | 0.630 | | 0.667 | 0.515 | 0.424 |
| Pseudo $R^2$ | | | 0.1467 | | | |

Columns (5) and (6) employ Propensity Score Matching (PSM) to mitigate possible selection bias in the formation of bank-firm relationships. Column (5) applies a treatment-on-the-treated analysis where greenwashing scores are set to zero when they are below zero, following a conservative assumption that such firms are unlikely to be engaged in symbolic ESG misrepresentation. Column (6) excludes firms suspected of "brownwashing"-i.e., firms with low ESG performance and low disclosure-ensuring that only potential greenwashers are retained. In both cases, the bank-firm relationship variable continues to exert a significantly positive effect (coefficients: 0.028 and 0.047, respectively), reinforcing the robustness of the main result even under varying sample definitions.

Across all six specifications, the results consistently support the hypothesis that closer bank-firm affiliations lead to a greater likelihood of ESG greenwashing. The estimated coefficients are positive and significant in all columns, regardless of whether the tests involve alternate greenwashing metrics, model adjustments (Tobit), or sample restrictions (PSM, brown-washing exclusion). Standard errors are clustered at the industry and year levels, and t-statistics are reported in parentheses. Significance levels are denoted as follows: ***, **, and * correspond to the 1%, 5%, and 10% thresholds, respectively.

To account for firms' external borrowing behavior, we construct a borrowing index using the natural logarithm of the sum of entrusted loans, entrusted wealth management products, and informal lending amounts, plus one. Given the concealed nature of informal lending, we follow Jiang et al. (2010) and use "other receivables" as a proxy for private borrowing. We re-estimate the baseline regression by including this firm-level borrowing index as an additional control variable. The results, presented in Table 6, show that the coefficient on RS (bank-firm relationship) remains positive and statistically significant at the 1% level (coefficient = 0.054), indicating that the effect of bank-firm ties on greenwashing persists even after controlling for firms' borrowing activity. This finding is consistent with our baseline results and further confirms the robustness of the relationship.

To further validate the robustness of the baseline findings and rule out the possibility that the observed differences between the treatment and control groups are driven by random factors, we conduct a placebo test using simulated treatment groups. Specifically, we randomly assign a subset of firms as "pseudo-treated" and re-estimate the regression using greenwashing as the dependent variable. If the estimated coefficients from these randomized models remain significant, it would suggest that the original results might be driven by spurious correlations or unobserved random shocks.

To ensure robustness, we repeat this procedure 500 times, each time randomly selecting firms to construct the placebo treatment group. Fig 1 presents the kernel density distributions of the estimated coefficients and corresponding p-values across these 500 simulations.

The results clearly show that the distribution of placebo coefficients deviates substantially from the baseline coefficient, and that more than 90% of the simulated coefficients are statistically insignificant at the 10% level ($p > 0.1$). Furthermore, none of the placebo estimates closely approximate the magnitude of the baseline estimate. These findings provide strong evidence that the main regression results are unlikely to be driven by random chance, and support the statistical robustness and reliability of the baseline specification.

To address potential endogeneity issues arising from omitted variables and reverse causality, this study adopts an instrumental variable (IV) approach. Specifically, we utilize the average bank-firm relationship intensity of firms in the same industry and city as the instrument. The rationale is that a firm's likelihood of establishing banking ties is shaped by the local financial development and regulatory environment, which vary across regions and industries. This instrumental variable is expected to be strongly correlated with a firm's own bank affiliation status (relevance condition), while it is unlikely to directly affect the firm's greenwashing behavior other than through its influence on bank-firm ties (exogeneity condition).Table 7shows the two-stage least squares (2SLS) estimation. In the first stage, the IV significantly predicts bank-firm relationships (coefficient = 0.954, $p < 0.01$). The Kleibergen-Paap LM statistic rejects underidentification, and the Cragg-Donald Wald F-statistic exceeds the Stock-Yogo critical threshold, confirming instrument strength and relevance.

In the second stage, the estimated coefficient on bank-firm relationship (RS) remains significantly positive (coefficient = 0.061, $p < 0.01$), consistent with our baseline regression. These findings support the robustness of our main results and alleviate concerns of endogeneity bias.

Table 8 explores the heterogeneity of the effect of bank-firm relationships on greenwashing by analyzing firm ownership types and regional financial development. Columns (1) and (2) compare state-owned enterprises (SOEs) with non-state-owned enterprises (non-SOEs), revealing a stronger positive association between bank-firm relationships and greenwashing among non-SOEs. This suggests that privately owned firms may face heightened pressure to meet bank expectations through symbolic ESG practices, potentially due to their lack of regulatory privileges or more competitive market pressures compared to SOEs.

**Table 6. Robustness test of regression results controlling for firm-level borrowing indicators.**

| | (1) ESGW1 |
|---|---|
| RS | 0.054*** |
| | (3.544) |
| Size | −0.400*** |
| | (−15.898) |
| ROA | −0.665*** |
| | (−9.021) |
| Growth | 0.084*** |
| | (3.067) |
| Lev | 0.871*** |
| | (17.361) |
| FAR | 0.032 |
| | (0.387) |
| Indep | −1.322*** |
| | (−2.954) |
| Board | −0.145 |
| | (−1.262) |
| FirmAge | 0.043 |
| | (0.513) |
| BM | 0.312*** |
| | (6.367) |
| OCF | 0.092 |
| | (0.907) |
| Inst | 0.088 |
| | (0.945) |
| DC | −0.012*** |
| | (−4.246) |
| _cons | 9.068*** |
| | (26.197) |
| Firm FE | Yes |
| Industry*Year FE | Yes |
| N | 37512 |
| $R^2$ | 0.712 |
| Adj. $R^2$ | 0.674 |

Further, Columns (3) and (4) assess the moderating effect of regional financial agglomeration on the relationship between bank-firm ties and greenwashing. The results show that in regions with lower levels of financial agglomeration, firms with close banking relationships are significantly more likely to engage in greenwashing compared to firms in highly agglomerated financial centers.

One possible explanation is that in less financially developed regions, banks possess stronger bargaining power, and firms tend to be more dependent on bank financing due to limited access to alternative capital sources. This dependency may lead banks to relax environmental oversight standards, as their financial interests become intertwined with the firm's performance. Consequently, firms are incentivized to adopt symbolic ESG disclosures that meet surface-level expectations without committing to substantive environmental improvement.

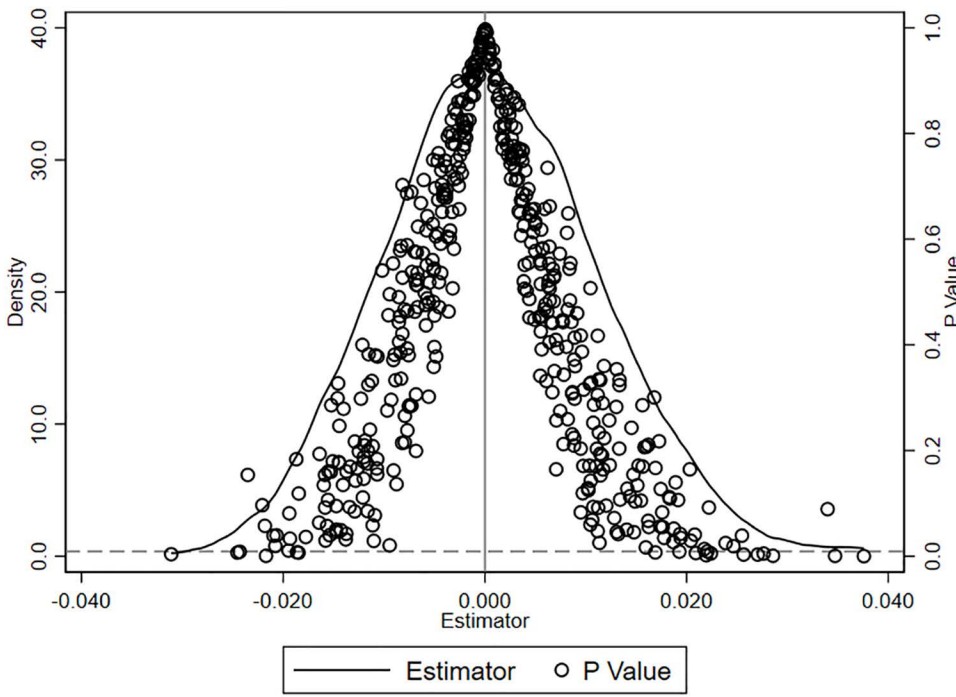

**Fig 1. Placebo test.**

In contrast, regions with high financial agglomeration typically feature more robust regulatory frameworks, stronger capital markets, and more active third-party oversight—including media scrutiny, environmental NGOs, and sustainability rating agencies. These mechanisms increase the cost and risk of greenwashing, thereby constraining firms from engaging in opportunistic ESG misrepresentation.

Moreover, in low-agglomeration regions, regulatory enforcement tends to be weaker, and the lack of independent monitoring bodies makes it easier for firms to obscure environmental shortcomings through close ties with banks. Firms in these regions often lack diversified financing options and therefore rely more heavily on their relationships with banks, which, in the absence of intense interbank competition, may reduce the stringency of environmental review processes. As a result, symbolic ESG compliance becomes more prevalent.

These findings highlight how regional financial development conditions can either constrain or amplify the greenwashing incentives associated with bank-firm relationships. In underdeveloped financial environments, such relationships may inadvertently encourage symbolic disclosure strategies by lowering the barriers to regulatory avoidance.

Finally, Columns (5) and (6) of the heterogeneity analysis examine the moderating effect of industry competition, measured by the Herfindahl-Hirschman Index (HHI), which captures market concentration. The results indicate that in industries with lower competition, firms with strong banking relationships are more likely to engage in greenwashing compared to those in more competitive sectors. This pattern can be attributed to two primary factors.

First, in less competitive markets, firms face reduced pressure from rivals to improve their actual ESG performance, thereby weakening the incentives for substantive environmental investment. Second, monopolistic or oligopolistic market structures create lower external pressure and higher stability, which may motivate firms to pursue short-term strategies-such as symbolic ESG disclosure-rather than genuine sustainability initiatives.

Close bank-firm ties may facilitate access to financing, enabling firms to obscure environmental problems or qualify for green finance without making substantive changes. Moreover, in concentrated markets, firms are subject to less investor

**Table 7. Endogeneity Test.**

| | (1) RS | (2) ESGW1 |
|---|---|---|
| iv | 0.954*** | |
| | (176.526) | |
| RS | | 0.061*** |
| | | (3.870) |
| Size | −0.003 | −0.412*** |
| | (−0.935) | (−16.309) |
| ROA | 0.019 | −0.655*** |
| | (0.664) | (−8.926) |
| Growth | 0.000 | 0.084*** |
| | (0.000) | (3.075) |
| Lev | 0.045** | 0.875*** |
| | (2.389) | (17.583) |
| FAR | 0.003 | 0.049 |
| | (0.175) | (0.588) |
| Indep | 0.021 | −1.324*** |
| | (0.436) | (−2.970) |
| Board | 0.078*** | −0.147 |
| | (4.602) | (−1.317) |
| FirmAge | −0.044 | 0.032 |
| | (−1.356) | (0.375) |
| BM | 0.017 | 0.312*** |
| | (1.621) | (6.352) |
| OCF | −0.010 | 0.100 |
| | (−0.535) | (1.024) |
| Inst | 0.055*** | 0.090 |
| | (3.068) | (0.975) |
| Firm FE | Yes | Yes |
| Industry*Year FE | Yes | Yes |
| N | 37529 | 37529 |
| $R^2$/Centered $R^2$ | 0.530 | 0.0472 |
| Adj. $R^2$/Uncentered $R^2$ | 0.473 | 0.0472 |

and consumer scrutiny, further reducing the reputational risk of greenwashing. The support of affiliated banks can also ease credit constraints, even when firms lack meaningful environmental actions, thereby reinforcing the tendency toward symbolic compliance.

In such settings, firms often face higher entry barriers and benefit from entrenched market positions, making them more inclined to maintain the status quo rather than invest in long-term ESG transformation. Tight banking relationships can reinforce this behavior by providing financial leverage to enhance the appearance of sustainability-e.g., through selective disclosure or ESG-themed financial packaging—rather than promoting substantive change.

This heterogeneity analysis underscores the importance of contextual factors-including ownership structure, regional financial development, and industry competition-in shaping how banking relationships influence greenwashing behavior. Firms operating in more challenging institutional and market environments appear more likely to adopt symbolic ESG strategies when backed by strong banking ties.

**Table 8. Heterogeneity.**

| | (1) | (2) | (3) | (4) | (5) | 6) |
|---|---|---|---|---|---|---|
| | SOE = 1 | SOE = 0 | FA>Median | FA<=Median | HHI>Median | HHI<=Median |
| Banking-frim relationship | 0.030 | 0.054*** | 0.034* | 0.071** | 0.055* | 0.062** |
| | (1.405) | (3.485) | (1.886) | (2.416) | (1.840) | (2.145) |
| Size | −0.421*** | −0.423*** | −0.441*** | −0.353*** | −0.359*** | −0.432*** |
| | (−9.626) | (−19.844) | (−9.791) | (−14.007) | (−13.251) | (−14.676) |
| ROA | 0.066 | −0.707*** | −0.525*** | −0.554*** | −0.656*** | −0.362*** |
| | (0.215) | (−4.851) | (−4.664) | (−4.908) | (−6.454) | (−3.567) |
| Growth | 0.099*** | 0.074** | 0.096*** | 0.074** | 0.073** | 0.076** |
| | (2.938) | (2.805) | (2.931) | (2.811) | (2.300) | (2.274) |
| Lev | 0.734*** | 0.920*** | 0.830*** | 0.844*** | 0.693*** | 0.906*** |
| | (7.193) | (21.547) | (11.794) | (16.192) | (6.945) | (17.119) |
| FAR | 0.044 | 0.119 | 0.027 | 0.080 | −0.003 | 0.079 |
| | (0.248) | (1.210) | (0.218) | (0.617) | (−0.020) | (1.530) |
| Indep | −1.580** | −1.089** | −1.433*** | −0.846 | −0.901* | −1.564** |
| | (−2.785) | (−2.571) | (−4.857) | (−1.288) | (−2.043) | (−2.792) |
| Board | −0.093 | −0.124 | −0.182** | −0.035 | −0.009 | −0.252* |
| | (−0.380) | (−1.499) | (−2.737) | (−0.172) | (−0.079) | (−2.043) |
| FirmAge | −0.059 | 0.022 | 0.104 | −0.068 | 0.186 | 0.005 |
| | (−0.250) | (0.255) | (0.735) | (−0.579) | (1.116) | (0.027) |
| BM | 0.206* | 0.416*** | 0.389*** | 0.249*** | 0.237*** | 0.386*** |
| | (2.004) | (7.259) | (4.099) | (4.632) | (4.776) | (4.623) |
| OCF | 0.016 | 0.123 | 0.009 | 0.152 | 0.084 | 0.087 |
| | (0.076) | (1.565) | (0.042) | (1.712) | (0.548) | (0.599) |
| Inst | 0.025 | 0.112 | 0.083 | −0.022 | −0.047 | 0.125 |
| | (0.211) | (1.094) | (0.852) | (−0.161) | (−0.656) | (1.139) |
| _cons | 9.775*** | 9.171*** | 9.722*** | 7.765*** | 7.239*** | 9.911*** |
| | (10.422) | (19.663) | (11.137) | (9.846) | (11.472) | (16.130) |
| Firm FE | Yes | Yes | Yes | Yes | Yes | Yes |
| Industry* | | | | | | |
| Year FE | Yes | Yes | Yes | Yes | Yes | Yes |
| N | 13385 | 24063 | 17952 | 18618 | 17303 | 18998 |
| $R^2$ | 0.737 | 0.717 | 0.740 | 0.729 | 0.733 | 0.739 |
| Adj. $R^2$ | 0.703 | 0.670 | 0.691 | 0.677 | 0.684 | 0.691 |

t-Statistics calculated using standard errors clustered by industry and year are reported in parentheses. ***, **, and * denote statistical significance at the 1%, 5%, and 10% levels.

This study posits that different dimensions of bank-firm relationships-including firms' equity holdings in banks, banks' equity ownership in firms, and executives' banking backgrounds—may exert varying effects on corporate greenwashing behavior. To explore these heterogeneous influences, we decompose the composite bank-firm relationship variable into three components:

CB (Corporate-to-Bank Equity Holding): A binary indicator equal to 1 if the firm holds equity in a commercial bank, and 0 otherwise.

BC (Bank-to-Corporate Equity Holding): A binary indicator equal to 1 if a commercial bank holds equity in the firm, and 0 otherwise.

MB (Managerial Banking Background): A binary indicator equal to 1 if at least one executive has previous employment experience in the banking sector, and 0 otherwise.

The regression results (see Table 9) show notable differences across these dimensions. Specifically, CB does not exhibit a statistically significant impact on greenwashing, suggesting that firms' ownership of bank equity may not meaningfully alter their ESG disclosure behavior. In contrast, both BC and MB demonstrate significant positive effects on greenwashing, indicating that these channels are more influential in promoting symbolic ESG compliance.

One possible explanation is that firms holding bank equity may typically do so at relatively low ownership levels, thereby lacking sufficient control or influence over the banks' behavior or lending preferences. As a result, bank favorability or oversight standards may not be materially altered, and no strong incentives for greenwashing are observed.

Conversely, when banks hold equity stakes in firms, this may give rise to closely aligned financial interests, creating implicit pressure on the firm to maintain favorable performance indicators, including ESG credentials. Banks in this position may prioritize financial returns over environmental integrity, encouraging firms to engage in greenwashing to improve market valuation or facilitate access to capital. Furthermore, banks may lower environmental scrutiny or due diligence standards for firms in which they hold shares, due to conflicts of interest, thus exacerbating regulatory gaps and enabling symbolic ESG disclosure.

Executives with banking backgrounds may also play a role by leveraging personal networks and familiarity with bank operations. These individuals may gain easier access to loans, reducing the need to pursue genuine ESG projects to attract financing. They may also use their influence to soften external oversight or bypass environmental review processes, especially if informal ties with banking institutions exist. In addition, such executives are likely more adept at understanding or circumventing ESG evaluation systems, thereby increasing the likelihood of sophisticated greenwashing strategies.

These findings highlight that the influence of bank-firm relationships on ESG behavior is not homogeneous but depends critically on the type and direction of the relationship, as well as the embedded institutional or personal connections. This nuanced understanding provides deeper insights into the mechanisms through which financial affiliations shape corporate sustainability practices.

## 5. Research findings and implications

### 5.1 Research findings

This study examines how non-lending bank-firm relationships-specifically bank equity ownership and executives with banking backgrounds-influence corporate greenwashing in the Chinese context. The findings reveal that firms with stronger banking affiliations are more likely to overstate their ESG performance, engaging in symbolic rather than substantive ESG disclosure. This aligns with prior work on symbolic compliance under institutional pressure (DiMaggio & Powell, 1983 [20]; Delmas & Burbano, 2011 [5]), and further demonstrates how banks, as institutional stakeholders, may enable such behavior through indirect influence. In contrast to the conventional view that banks enhance governance via monitoring (Diamond, 1984 [9]; Mahrt-Smith, 2006 [10]), our results suggest that in emerging markets characterized by weak regulatory enforcement and high information asymmetry, bank-firm relationships can incentivize reputational signaling at the expense of real ESG engagement (Weinstein & Yafeh [11], 1998; Luo et al., 2011 [12]).

Moreover, we identify executive compensation as a key mediating mechanism in the relationship between bank affiliations and greenwashing. Firms with banking ties are more likely to link executive bonuses and performance evaluations to ESG ratings, which, while intended to promote sustainability, may in fact incentivize opportunistic disclosure behavior. This supports insights from Agency Theory, which holds that in the presence of information asymmetry, agents (managers) may exploit symbolic metrics to meet principal (investor) expectations without delivering real performance (Jensen & Meckling, 1976 [40]; Kim & Lyon, 2015 [4]). The misalignment between disclosed ESG achievements and actual environmental or social improvements represents a subtle but powerful form of greenwashing.

**Table 9. Heterogeneity analysis of dimensions of bank-firm relationships.**

| | (1) ESGW1 | (2) ESGW1 | (3) ESGW1 |
|---|---|---|---|
| CB | −0.017 | | |
| | (−0.490) | | |
| BC | | 0.187** | |
| | | (2.148) | |
| MB | | | 0.046*** |
| | | | (3.342) |
| Size | −0.412*** | −0.411*** | −0.412*** |
| | (−16.866) | (−16.596) | (−16.383) |
| ROA | −0.657*** | −0.656*** | −0.655*** |
| | (−9.021) | (−8.831) | (−8.969) |
| Growth | 0.085*** | 0.085*** | 0.084*** |
| | (3.143) | (3.141) | (3.085) |
| Lev | 0.878*** | 0.882*** | 0.875*** |
| | (17.593) | (17.597) | (17.532) |
| FAR | 0.046 | 0.042 | 0.048 |
| | (0.562) | (0.522) | (0.583) |
| Indep | −1.317*** | −1.317*** | −1.321*** |
| | (−2.975) | (−2.987) | (−2.964) |
| Board | −0.137 | −0.138 | −0.145 |
| | (−1.221) | (−1.222) | (−1.282) |
| FirmAge | 0.036 | 0.026 | 0.032 |
| | (0.423) | (0.301) | (0.378) |
| BM | 0.311*** | 0.316*** | 0.312*** |
| | (6.443) | (6.361) | (6.326) |
| OCF | 0.099 | 0.104 | 0.099 |
| | (1.013) | (1.053) | (1.012) |
| Inst | 0.098 | 0.097 | 0.092 |
| | (1.080) | (1.072) | (0.994) |
| _cons | 9.131*** | 9.127*** | 9.146*** |
| | (27.134) | (26.201) | (26.336) |
| Firm FE | Yes | Yes | Yes |
| Industry*Year FE | Yes | Yes | Yes |
| N | 37529 | 37529 | 37529 |
| $R^2$ | 0.712 | 0.712 | 0.712 |
| Adj. $R^2$ | 0.674 | 0.674 | 0.674 |

In addition, we find that financialization-measured by the ratio of financial assets to total assets-exacerbates the likelihood of greenwashing. Financialized firms are more focused on short-term capital market returns and thus more prone to using ESG disclosure as a market-facing strategy. This finding is consistent with prior studies showing that financialization can dilute firms' commitment to long-term sustainability (Lin et al., 2015 [13]; Davis, 2009 [41]). On the other hand, firms that adopt a positive and transparent tone in their ESG narratives, as captured via textual analysis of annual reports, exhibit lower tendencies to greenwash. This aligns with Zhang (2022) [6] and Ge et al. (2023) [42], who suggest that disclosure tone reflects organizational transparency and can mitigate the risk of symbolic

compliance. The dual role of financial strategy and communication framing thus plays a central part in shaping ESG reporting behavior.

Finally, our heterogeneity analysis reveals that the impact of bank-firm relationships on greenwashing varies significantly across ownership types and regional financial environments. Non-state-owned enterprises (non-SOEs) are more vulnerable to greenwashing under banking influence compared to state-owned enterprises (SOEs), likely due to greater external pressure and weaker formal oversight (Agarwal & Elston, 2001 [17]; Pan & Tian, 2014 [43]). Likewise, firms located in regions with lower financial agglomeration-often characterized by limited institutional infrastructure-display higher greenwashing tendencies. These results suggest that both internal firm structures and external institutional contexts shape the effectiveness and outcomes of banking relationships in the ESG domain.

Robustness checks using alternative greenwashing measures, fixed-effects regressions, and falsification tests (e.g., placebo models) support the consistency of these findings. While we acknowledge the possibility of residual endogeneity, especially in observational data, the convergence of results across different model specifications enhances confidence in the causal mechanisms proposed.

## 5.2 Policy, managerial, and theoretical implications

The findings of this study have important implications for regulators, financial institutions, and corporate decision-makers seeking to curb ESG greenwashing and promote authentic sustainability practices.

From a regulatory perspective, our findings call for more precise and enforceable ESG disclosure standards, especially for bank-affiliated firms. Rather than general disclosure mandates, regulators should require firms to provide verifiable, consistent, and disaggregated ESG data, including clear differentiation between inputs, outputs, and outcomes of sustainability efforts (Busch et al., 2020 [44]). Particular emphasis should be placed on the alignment between ESG disclosures and executive compensation schemes, which currently lack transparency. Regulatory agencies should also develop industry-specific ESG indicators, mandate third-party assurance of ESG data, and promote greater regional oversight in areas with weaker financial ecosystems. These steps would help limit the capacity of firms to exploit symbolic ESG language and create accountability mechanisms that reward real impact over superficial compliance (Deegan, 2022 [45]).

For financial institutions, especially commercial banks and asset managers, the results suggest a need to reassess ESG risk evaluation frameworks. ESG rating inflation-particularly in bank-connected firms-may mislead investors and jeopardize long-term portfolio stability. Institutions should not rely solely on ESG ratings, but rather incorporate performance-to-disclosure alignment checks, independent ESG audits, and governance structure assessments into their due diligence processes (Flammer, 2015 [46]; Liang & Renneboog, 2017 [47]). Moreover, financial institutions should critically evaluate their incentive systems and equity holding structures, ensuring they do not unintentionally reward symbolic ESG behavior. Embedding long-term ESG metrics into investment screening and performance evaluation can help financial intermediaries move from ESG rhetoric to ESG responsibility.

Corporate managers, particularly those in non-SOEs and underdeveloped financial regions, should recognize the reputational and strategic risks associated with greenwashing. Beyond legal compliance, managers must focus on building internal ESG governance capacity, including dedicated sustainability committees, ESG performance audits, and stakeholder engagement mechanisms (Hsu et al., 2018) [48]. The study also underscores the need to reform executive incentive schemes, shifting from disclosure-based rewards to impact-based benchmarks (e.g., emission reductions, workforce diversity outcomes). In terms of disclosure strategy, firms should avoid excessive ESG optimism and adopt a balanced narrative tone to maintain credibility and stakeholder trust (Ge et al., 2023) [42].

Theoretically, this study contributes to the ESG and corporate governance literature by showing how Institutional, Agency, and Legitimacy Theories intersect to explain symbolic ESG behavior in emerging market contexts. While institutional pressures promote ESG disclosure, agency problems within financialized governance systems allow managers to manipulate such disclosures. Meanwhile, legitimacy considerations drive firms to conform to ESG expectations even in

the absence of real change. Our analysis of mechanisms (executive incentives) and moderators (financialization and tone) expands these frameworks by demonstrating how symbolic compliance can be institutionally sustained and strategically rationalized.

Future research could strengthen causal inference by leveraging exogenous shocks (e.g., green finance regulations, ESG index inclusion) or applying machine learning techniques to detect subtle forms of greenwashing in ESG texts and narratives (Serafeim, 2020) [49]. Comparative studies across countries with different ESG institutional frameworks would also help generalize the findings.

## Supporting information

**S1 Dataset.**
(XLSX)

## Author contributions

**Conceptualization:** Hongyu Liu.

**Data curation:** Hongyu Liu.

**Formal analysis:** Hongyu Liu.

**Funding acquisition:** Hongyu Liu.

**Investigation:** Hongyu Liu.

**Methodology:** Hongyu Liu.

**Project administration:** Hongyu Liu.

**Resources:** Hongyu Liu.

**Software:** Hongyu Liu.

**Supervision:** Hongyu Liu.

**Validation:** Hongyu Liu.

**Visualization:** Hongyu Liu.

**Writing – original draft:** Hongyu Liu.

**Writing – review & editing:** Hongyu Liu.

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
