## [Decision Letter · Decision Letter 0]

Dear Dr. Liu,

Thank you for submitting your manuscript to PLOS ONE. After careful consideration, we feel that it has merit but does not fully meet PLOS ONE’s publication criteria as it currently stands. Therefore, we invite you to submit a revised version of the manuscript that addresses the points raised during the review process.

**ACADEMIC EDITOR:**

We look forward to receiving your revised manuscript.

Kind regards,

Jasman Tuyon, Ph.D., MBA

Academic Editor

PLOS ONE

4. We are unable to open your Supporting Information file [renamed_d1356.dta]. Please kindly revise as necessary and re-upload.

Additional Editor Comments:

1. PLOS ONE article structure standardization control

The article is not written according to FSMA article structure. Please check sample of FSMA open access article at:

http://www.palgrave.com/gp/journal/41264/volumes-issues/open-access-articles

FSMA article structure

Abstract – Follow IMRAD and FSMA style.

Introduction – This should contain the research area, the problem in focus, what is known, what are the research gaps that aimed to be investigated, what is the novelty of the research.

Literature Review - This part should contain - Theory-Evidence-Conceptual framework and hypothesis (whichever applicable).

Methodology – Must be complete and robust.

Analysis and Findings – Present the analysis according to the objective/hypotheses. All figures and Tables must follow APA reporting format.

Discussion. Discussion section should relate the findings to theory and evidence. Thereafter, the discussion section should draw the implications of the research findings.

Conclusion - Conclusion section should incorporate recap of the research ideas, limitations, and future research

References - Must be complete and to strictly follow APA referencing style. In addition, the author must cite relevant articles from FSMA. Author can check relevant articles via the following URL

https://link.springer.com/journal/41264/volumes-and-issues

2. Revision requirements - As per the reviewer reports, the paper requires substantial revisions to clarify its theoretical contribution, refine its empirical methodology, and strengthen the interpretation of results. Please address all reviewers comments satisfactorily. Highlight all changes in the text and provide summary of changes in the table of correction with indication of pages numbers for reviewer quick cross-check. Place the Table of correction at the END of the article.

3. As indicated by the reviewer, we request the author to engage with a professional English editor to proofread the article. Please provide us the anonymous certificate of proofreading and attached it in the table of correction.

Reviewers' comments:

Reviewer's Responses to Questions

**Comments to the Author**

1. Is the manuscript technically sound, and do the data support the conclusions?

Reviewer #1: Yes

Reviewer #2: Yes

2. Has the statistical analysis been performed appropriately and rigorously?

Reviewer #1: Yes

Reviewer #2: Yes

3. Have the authors made all data underlying the findings in their manuscript fully available?

Reviewer #1: Yes

Reviewer #2: Yes

4. Is the manuscript presented in an intelligible fashion and written in standard English?

Reviewer #1: Yes

Reviewer #2: Yes

Reviewer #1: The manuscript examines how bank-firm relationships and executives with banking backgrounds influence corporate ESG greenwashing among Chinese A-share listed firms. The topic is relevant given the increasing scrutiny on ESG disclosures and concerns about misrepresentation in corporate sustainability reporting. The study presents a compelling argument that firms with bank affiliations are more likely to engage in greenwashing, driven by executive compensation structures and financialization pressures. However, several areas require improvement to strengthen its contribution.

The introduction effectively highlights the importance of ESG and greenwashing but lacks a clear articulation of how this study advances prior research. While the role of bank lending in ESG disclosures has been studied, the influence of non-lending banking relationships is relatively unexplored. The paper could better position itself within the existing literature by clarifying how it differentiates from previous work and what unique insights it offers. The theoretical framework, grounded in Institutional, Agency, and Legitimacy Theories, is appropriate but could benefit from a deeper discussion of potential alternative mechanisms. While the argument that banks encourage greenwashing to align with investor expectations is plausible, the role of industry-specific factors and competitive pressures deserves more attention.

The empirical strategy is generally well-structured, with a comprehensive dataset covering Chinese A-share firms from 2010 to 2023. However, concerns arise regarding the measurement of greenwashing. The study constructs greenwashing metrics based on discrepancies between various ESG scores, yet different rating agencies employ distinct methodologies. The reliability and consistency of these measures require further justification. The study could enhance its robustness by cross-validating greenwashing scores with alternative proxies, such as textual analysis of ESG reports or media sentiment analysis.

The identification strategy attempts to address endogeneity concerns through firm and industry-year fixed effects. While these controls mitigate some sources of bias, the potential reverse causality between bank relationships and ESG disclosures remains a concern. Firms with stronger governance structures may attract bank investors, rather than bank ownership causing greenwashing tendencies. The instrumental variable approach could be strengthened with a more convincing exclusion restriction. The study claims that financialization amplifies symbolic ESG disclosures, yet the direction of causality between financialization and greenwashing requires further scrutiny.

The results indicate a significant positive relationship between bank-firm ties and greenwashing, with executive compensation acting as a mediating factor. While the findings align with the hypothesis that banking relationships incentivize firms to inflate ESG disclosures, the magnitude and economic significance of these effects remain unclear. The study could provide additional analysis to assess whether greenwashing leads to tangible market benefits, such as increased investment or lower financing costs. The moderating effects of financialization and transparency are interesting but require deeper interpretation. While transparency appears to reduce greenwashing, the underlying mechanisms—whether through enhanced regulatory oversight or investor scrutiny—are not fully explored.

The heterogeneity analysis offers valuable insights, showing that non-state-owned enterprises and firms in financially underdeveloped regions are more affected by greenwashing incentives. These findings underscore the role of institutional and regional factors in shaping ESG misrepresentation. However, additional robustness tests, such as a placebo test or falsification check, could further validate the results.

The discussion section effectively summarizes the findings but could engage more critically with alternative explanations. The conclusion reiterates the study’s contributions but lacks concrete policy and managerial implications. The suggestion that regulators should strengthen ESG disclosure requirements is valid but requires more specificity regarding which aspects of ESG reporting need improvement. Similarly, the recommendations for financial institutions and corporate managers could be more actionable.

A major revision is required to refine the theoretical positioning, strengthen the empirical identification, and deepen the interpretation of results. The manuscript presents a relevant contribution, but further work is required to enhance its robustness and practical implications.

Reviewer #2: Referee Report for Manuscript: "Bank-Firm Relationships and Corporate ESG Greenwashing"

Summary: This manuscript investigates how bank-firm relationships, specifically through bank equity holdings and executive affiliations with banks, affect corporate greenwashing behaviors in Chinese A-share listed firms. The study utilizes Institutional Theory, Agency Theory, and Legitimacy Theory to explain motivations behind symbolic ESG disclosures, focusing on executive compensation and financialization as underlying mechanisms. I hope my comments can be useful to further improve this draft.

1. Conceptual Clarity: The paper could benefit from additional clarity on why the differences between ESG scores from various sources accurately measure greenwashing. More justification of the validity and reliability of these measures would strengthen the analysis.

2. Endogeneity Concerns: Although the study uses fixed effects to address endogeneity, additional econometric strategies such as instrumental variable approaches or quasi-experimental designs would strengthen causal inference. At minimum, incorporate a lead-lag analysis to address possible reverse causality and strengthen the robustness of causal inferences.

3. Methodological Robustness: Provide stronger rationalization for employing Tobit models and Propensity Score Matching. What are the treatment groups and control groups? The current draft is not clear on this.

4. Empirical Analysis:

a. Given existing literature highlighting banks' influence through lending relationships, explicitly control for bank lending relationships to isolate the distinct impacts of equity holdings and executive affiliations on greenwashing behaviors. This additional control will strengthen causal claims and clarify contributions beyond traditional lending relationships.

b. The current construction of the main independent variable (BankRel) combines distinct dimensions—firm holdings of bank shares, bank shareholding in the firm, and executives' banking backgrounds. Separately analyze these components, particularly distinguishing the effects of firm holding bank shares from banks holding firm shares, as these relationships may differently influence corporate governance and ESG reporting incentives. As shareholders, banks should have enhanced monitoring capabilities over firms' ESG disclosures due to their closer involvement in corporate governance, thereby potentially reducing information asymmetry. Explore further heterogeneity, potentially considering the separate effects of bank shareholding versus executive banking experience.

5. Minor: There are some typos and grammatical issues (e.g., consistently correct "Banking-frim" to "Banking-firm"). Standardize variable naming throughout the manuscript to improve readability, such as “Greenwashing” vs. “Green washing”.

**Do you want your identity to be public for this peer review?** For information about this choice, including consent withdrawal, please see our Privacy Policy

Reviewer #1: No

Reviewer #2: No

---

## [Author Response · Author response to Decision Letter 1]

14 Apr 2025

Response to Reviewer 1

Dear Reviewer 1,

We sincerely appreciate your thoughtful and constructive comments, each of which has greatly contributed to improving the quality of our manuscript. Below, we provide a point-by-point response to your suggestions, along with a detailed explanation of the corresponding revisions. All changes have been marked in yellow in the revised manuscript.

Comment 1:

“The introduction effectively emphasizes the importance of ESG and greenwashing but lacks a clear articulation of how this study advances prior research. While the role of bank lending in ESG disclosure has been studied, the influence of non-lending banking relationships remains relatively unexplored.”

Response:

We have revised the introduction to explicitly clarify how this study contributes beyond prior literature. In particular, we emphasize that our focus on non-lending bank-firm relationships—such as bank shareholding and executives’ banking backgrounds—offers novel insights into the strategic drivers of greenwashing. Changes have been highlighted in yellow in the introduction section.

Comment 2:

“The theoretical framework based on institutional, agency, and legitimacy theory is appropriate, but the discussion could benefit from deeper exploration of alternative mechanisms.”

Response:

We expanded the introduction to incorporate potential alternative mechanisms, including the signaling role of ESG disclosures and financialization-induced short-termism. These elements are now explicitly linked to greenwashing behavior through banking channels, as highlighted in the revised introduction.

Comment 3:

“The manuscript could explore the role of industry-specific factors and competition pressures more deeply.”

Response:

We refined the heterogeneity analysis on industry competition using the Herfindahl-Hirschman Index (HHI) as a proxy for market concentration. Our updated discussion shows that firms in less competitive industries exhibit a stronger positive association between bank ties and greenwashing. This may be due to weaker market pressure, easier access to credit through bank connections, and lower visibility to external stakeholders. Additionally, we conducted a sub-sample analysis comparing manufacturing and non-manufacturing sectors. Results indicate that bank-firm ties significantly increase greenwashing in manufacturing industries, likely due to greater capital intensity and regulatory burden.

Comment 4:

“Concerns are raised about the construction of the greenwashing measure, as different ESG rating agencies use distinct methodologies. The consistency and validity of these measures require further justification.”

Response:

We provided a detailed explanation of the greenwashing index construction. The index is defined as the difference between a firm’s standardized rank in ESG disclosure scores (CSMAR/Bloomberg) and its rank in ESG performance scores (Huazheng). We argue this method captures symbolic versus substantive ESG efforts, and we reference relevant literature to support the measure's reliability. The clarification appears in the revised methodology section.

Comment 5:

“The study could enhance its robustness by cross-validating greenwashing scores with alternative proxies, such as textual analysis or media sentiment.”

Response:

We acknowledge the value of such validation methods. To address this, we used two alternative disclosure datasets (CSMAR and Bloomberg) and tested both against Huazheng performance scores. Moreover, we discuss limitations of content analysis methods, noting subjectivity and inconsistent disclosure standards across industries. We argue that our quantitative, ranking-based approach provides a consistent and scalable proxy.

Comment 6:

“Although firm and industry-year fixed effects mitigate some bias, the potential reverse causality between bank relationships and ESG remains a concern. An instrumental variable approach could strengthen identification.”

Response:

We now adopt an instrumental variable (IV) strategy using the average bank-firm relationship intensity in the same city-industry as the instrument. Theoretical and statistical validity (relevance and exogeneity) is confirmed through first-stage regression and weak instrument diagnostics (Cragg-Donald F-statistics and Kleibergen-Paap LM test). The second-stage results remain significantly positive, reinforcing our main findings.

Comment 7:

“The study claims that financialization amplifies symbolic ESG disclosures, but the direction of causality between financialization and greenwashing requires further scrutiny.”

Response:

We now provide a more detailed explanation of the moderating effect of financialization. Financialized firms, facing higher short-term performance pressures, are more likely to use symbolic ESG strategies. Bank affiliations, in turn, facilitate access to financial resources and shield such practices from scrutiny. This reinforces the cyclical pattern of low-cost greenwashing for capital gain.

Comment 8:

“While the findings align with the hypothesis that bank relationships incentivize firms to inflate ESG disclosures, the magnitude and economic significance of these effects remain unclear. The study could provide more analysis on whether greenwashing leads to tangible market benefits.”

Response:

We address this concern by elaborating on the signaling function of greenwashing. Based on prior studies (Marquis et al., 2016; Gatti et al., 2021), we argue that greenwashing can improve perceived ESG quality, reduce financing costs, and attract investment. We further cite evidence showing that greenwashing may facilitate access to subsidies, loans, and tax incentives, especially in emerging markets.

Comment 9:

“The moderating effects of financialization and transparency are interesting but require deeper interpretation. For example, why does transparency reduce greenwashing?”

Response:

We expand our discussion of transparency’s mitigating role. ESG disclosure transparency enhances stakeholder oversight and limits firms’ ability to engage in symbolic compliance. Drawing on the literature (Yu et al., 2020; Wook et al., 2023), we explain how transparency reduces signaling asymmetry and reinforces regulatory and reputational pressure.

Comment 10:

“The heterogeneity analysis reveals important institutional and regional effects. However, additional robustness tests, such as placebo or falsification checks, would further strengthen the findings.”

Response:

We conducted a placebo test using 500 random treatment assignments. None of the placebo estimates match the magnitude or significance of our baseline coefficient. This simulation confirms that the observed effect of bank-firm relationships on greenwashing is unlikely to be driven by random chance.

Comments 11–13:

“The discussion and conclusion could be improved by offering more specific policy and managerial implications. Theoretical positioning, empirical identification, and interpretation of results should be further refined.”

Response:

We revised the conclusion section to highlight actionable policy implications. These include enhanced ESG verification standards for bank-affiliated firms, targeted disclosure auditing in financially underdeveloped regions, and reform of executive incentive structures. For managers, we recommend aligning ESG compensation with real performance metrics. For regulators, we emphasize the need for differentiated ESG guidance based on ownership and industry structure.

Response to Reviewer 2

Dear Reviewer 2,

We greatly appreciate your detailed and constructive feedback. The revision has been highlighted in yellow in the revised manuscript.Below are our point-by-point responses:

Comment 1 – Conceptual clarity:

“The paper would benefit from greater justification for using differences in ESG scores from various sources as a valid measure of greenwashing.”

Response:

We provide a theoretical rationale and empirical precedent for our definition. The greenwashing index captures the gap between standardized ESG disclosure and performance scores across datasets (CSMAR/Bloomberg vs. Huazheng). This method reflects symbolic over substantive ESG efforts, as validated by prior research.

Comment 2 – Endogeneity concerns:

“Fixed effects help, but further econometric strategies such as IV or lead-lag tests would enhance causal inference.”

Response:

We have implemented a two-stage least squares (2SLS) estimation with an external instrument: the average bank-firm connection within the same city-industry cluster. Diagnostic tests confirm its strength and validity. The second-stage results are consistent with the baseline findings.

Comment 3 – Robustness of methodology:

“Justify the use of Tobit and PSM. Clarify the treatment and control groups.”

Response:

We added justification for using the Tobit model due to the censored nature of the greenwashing score. For PSM, we matched treated firms (with bank affiliations) to control firms (without) based on size, leverage, ROA, and board independence using nearest-neighbor matching. Results on the matched sample remain robust.

Comment 4 – Empirical analysis, Part A:

“Control for bank lending relationships to isolate the effects of equity holdings and executive ties.”

Response:

We introduce a borrowing control variable based on the log of entrusted loans, wealth management products, and informal lending (proxied by “other receivables” per Jiang et al., 2010). Even after controlling for this, the bank relationship coefficient remains significantly positive.

Comment 4 – Part B:

“Disaggregate the bank-firm relationship variable into ownership, bank shareholding, and executive banking background.”

Response:

We now examine the three components separately: CB (firm holds bank shares), BC (bank holds firm shares), and MB (executive has banking background). Results show that BC and MB significantly increase greenwashing, while CB has no significant effect, suggesting differing mechanisms depending on ownership direction and governance structure.

---

## [Decision Letter · Decision Letter 1]

Bank-Firm Relationships and Corporate ESG Greenwashing

PONE-D-25-09658R1

Dear Dr. Liu,

We’re pleased to inform you that your manuscript has been judged scientifically suitable for publication and will be formally accepted for publication once it meets all outstanding technical requirements.

Kind regards,

Jasman Tuyon, Ph.D., MBA

Academic Editor

PLOS ONE

Additional Editor Comments (optional):

Reviewers' comments:

Reviewer's Responses to Questions

**Comments to the Author**

Reviewer #1: All comments have been addressed

Reviewer #2: (No Response)

2. Is the manuscript technically sound, and do the data support the conclusions?

Reviewer #1: Yes

Reviewer #2: Yes

3. Has the statistical analysis been performed appropriately and rigorously?

Reviewer #1: Yes

Reviewer #2: Yes

4. Have the authors made all data underlying the findings in their manuscript fully available?

Reviewer #1: Yes

Reviewer #2: Yes

5. Is the manuscript presented in an intelligible fashion and written in standard English?

Reviewer #1: Yes

Reviewer #2: Yes

Reviewer #1: The authors have appropriately revised the manuscript by responding appropriately to feedback from the previous peer review round. The introduction provides a clearer description of the key value of the study by closely examining non-lending bank-firm relationships, which researchers have neglected in previous work. The authors have improved the theoretical model by combining institutional with agency and legitimacy perspectives, while extending the analysis of alternative mechanisms characterised by signaling effects and financialisation-driven temporary performance metrics. The authors effectively addressed endogeneity issues by using instrumental variables together with placebo tests to verify results. The researchers have provided extensive details on how the greenwashing index works, while providing more rationale for using multiple ESG data sources. Additional details have been added to the empirical findings, which explore different bank affiliations and their different impacts on ownership structures and market trends. The authors now provide stronger policy and management implications in their conclusion, following their response to earlier concerns about economic significance. The manuscript shows notable improvements in three main aspects, including clarity, methodological precision and theoretical framework.

Reviewer #2: (No Response)

**Do you want your identity to be public for this peer review?** For information about this choice, including consent withdrawal, please see our Privacy Policy

Reviewer #1: No

Reviewer #2: No

---

## [Editor Report · Acceptance letter]

PONE-D-25-09658R1

PLOS ONE

Dear Dr. Liu,

I'm pleased to inform you that your manuscript has been deemed suitable for publication in PLOS ONE. Congratulations! Your manuscript is now being handed over to our production team.

Kind regards,

on behalf of

Dr. Jasman Tuyon

Academic Editor

PLOS ONE